# HIV self-testing awareness among African refugee male sex workers in Italy: A mixed-methods study

Henry Delali Dakpui[1], Gamji Rabiu Abu-Ba'are[2]*, Mubarik Sena Saaka[1], Donte Boyd[3,4], Orlando Harris[5,6], Giovanni Zardini[7]

1 Behavioral, Sexual, and Global Health Lab, Accra, Ghana, 2 School of Nursing, University of Rochester, Rochester, New York, United States of America, 3 College of Social Work, The Ohio State University, Columbus, Ohio, United States of America, 4 Center for Interdisciplinary Research on AIDS (CIRA), Yale University, New Haven, Connecticut, United States of America, 5 Center for AIDS Prevention Studies (CAPS), Division of Prevention Science, Department of Medicine, University of California, San Francisco, California, United States of America, 6 Department of Community Health Systems, School of Nursing, University of California, San Francisco, California, United States of America, 7 Pink Refugees, Verona, Italy

* GamjiRabiu_Abubaare@urmc.rochester.edu

**Editor:** Serge Tonen-Wolyec, University of Bunia; Centre Interdisciplinaire de Recherche Translationnelle en Medecine et Sciences de la Sante (CIRTMSS), CONGO, THE DEMOCRATIC REPUBLIC OF THE

## Abstract

HIV disproportionately affects African refugee male sex workers (ARMSWs) in Italy, who face individual, structural and systemic barriers to HIV prevention and care services. HIV self-testing (HIVST) offers a promising strategy to improve testing access, yet awareness remains understudied in this population. This study examines HIVST awareness and associated factors among ARMSWs to inform targeted interventions. A mixed-methods sequential exploratory design was employed, combining quantitative surveys (n = 150) with qualitative interviews (20 in-depth interviews, 2 focus group discussions) among ARMSWs in Italy. Participants were recruited through venue-based and snowball sampling in partnership with a community organization. Quantitative data on HIVST awareness and correlates, including sociodemographic, healthcare access, and sex work characteristics, were analyzed using chi-square tests and logistic regression. Qualitative data from audio-recorded interviews underwent summative content analysis to explore awareness pathways and perceptions. Key findings revealed only 45% of participants were aware of HIVST, with just 47.8% of these having ever used a self-test. Higher education (aOR=1.92, p = 0.022) and prior STI testing (aOR=2.32, p = 0.015) significantly predicted awareness of HIVST. Qualitative data showed two awareness pathways: first-time exposure through the study and prior encounters via clients or community programs. Participants highlighted HIVST's privacy and convenience as key benefits. Community-based, peer-led approaches, combined with healthcare provider engagement, are essential for increasing HIVST awareness and uptake among ARMSWs. These findings have broader implications for improving HIV testing strategies among hard-to-reach migrant populations across Europe.

the Creative Commons Attribution License, which permits unrestricted use, distribution, and reproduction in any medium, provided the original author and source are credited.

**Data availability statement:** The dataset generated and analyzed during the current study cannot be publicly shared due to ethical restrictions imposed by the National Ethics Committee for Clinical Trials, Italian Ministry of Health, as it contains sensitive information from a highly vulnerable population that could compromise participant confidentiality even after de-identification. De-identified data may be made available upon reasonable request, subject to approval by the relevant ethics committee. Data access requests should be directed to the Scientific Secretariat of the National Ethics Committee (CEN) at the Istituto Superiore di Sanità (ISS) in Italy (email: segreteria.comitatoetico@iss.it).

**Funding:** The project described was supported by award number P30MH062294 from the National Institute of Mental Health (NIMH) through the Center for Interdisciplinary Research on AIDS (CIRA), Yale School of Public Health. The content is solely the responsibility of the authors and does not necessarily represent the official views of the Center for Interdisciplinary Research on AIDS, the National Institute of Mental Health, or the National Institutes of Health. The funders had no role in study design, data collection and analysis, decision to publish, or preparation of the manuscript.

**Competing interests:** The authors have declared that no competing interests exist.

## Introduction

The HIV epidemic continues to be a major global public health challenge, disproportionately affecting marginalized groups such as migrants, sex workers, and racial and sexual minorities [1–3]. In Europe, African migrants face persistent structural and social barriers to HIV prevention, testing, and treatment, including legal insecurity, stigma, and limited access to healthcare services [4,5]. These challenges are often intensified for African refugees, who frequently experience compounded vulnerabilities related to displacement, trauma, and social exclusion [6,7].

Male sex workers (MSWs), and particularly African refugee male sex workers (ARMSWs), remain largely overlooked in HIV research and programming in Europe, where attention has historically focused on female sex workers or men who have sex with men (MSM) more broadly [8–11]. ARMSWs operate at the intersection of migration, sex work, and sexual minority status, often facing stigma, economic precarity, and barriers to healthcare access that limit engagement with HIV testing services [8,9,14–16] Traumatic migration experiences and language barriers further exacerbate these challenges and contribute to delayed or foregone HIV testing [8,9].

In Italy, HIV testing services are delivered primarily through facility-based and community-based models, alongside HIV self-testing (HIVST), which has been legally available since 2016 [10,11]. While facility-based testing remains the dominant approach, many African refugees, particularly those who are undocumented, encounter significant obstacles to accessing these services [10–15] Community-based testing offers greater accessibility but remains limited in scale [11]. HIVST has emerged as a promising alternative by enabling private, autonomous testing and has been shown to increase testing uptake among key populations in diverse settings [12,14–16].

Despite the growing availability of HIVST in Europe, evidence on awareness and uptake remains limited and unevenly distributed across populations. Studies from Spain, the Netherlands, and Italy have documented persistently low awareness and use of HIVST among MSM and general populations, even several years after legal authorization, highlighting gaps in dissemination and access [17–23]. These studies report that HIVST is valued for its privacy and convenience, yet remains underutilized, particularly among individuals facing social and structural barriers to healthcare. Importantly, European HIVST research has largely focused on MSM or general users, with little attention to migrants, refugees, or sex workers as distinct groups.

To date, no published study has specifically examined awareness of HIVST among ARMSWs in Italy or elsewhere in Europe. This represents a critical gap in the literature, as ARMSWs face compounded barriers to HIV testing related to migration status, criminalization of sex work, economic precarity, stigma, and limited engagement with formal healthcare systems. Understanding how ARMSWs become aware of HIVST, and the factors that shape this awareness, is essential for designing inclusive, culturally responsive, and community-based HIV testing strategies. This mixed-methods study addresses this gap by examining levels of awareness of oral HIV self-testing among African refugee male sex workers in Italy and identifying sociodemographic, healthcare access, and sex work–related factors associated with

awareness. By integrating quantitative and qualitative data, we aim to generate contextually grounded evidence to inform HIV testing interventions tailored to the needs of this highly marginalized population.

## Materials and methods

### Study design

We employed a mixed-methods sequential exploratory design to investigate the awareness of HIVST among ARMSWs in Italy. The design combined an initial qualitative phase with a quantitative survey to enhance the depth and generalizability of the findings.

### Study setting

The study was conducted in two cities in Northern Italy; Verona and Torino, in partnership with Circolo Pink (Pink Refugees), a community-based nonprofit organization supporting LGBTQ+ refugees and migrants. Data collection was carried out in secure and accessible locations within their office spaces in Verona and a collaborating institution in Torino.

### Sample and sampling

The study population included adult ARMSWs currently living in Italy. Eligibility criteria were: (1) being a refugee from the Sub-Saharan Africa region, (2) residing in Italy, (3) aged 18 or older, (4) fluent in English or Italian, (5) self-identifying as a sex worker, and (6) having engaged in sex work within the last six months with someone of the same sex.

For the quantitative survey (n = 150), we used peer-led venue-based and snowball sampling strategy to recruite participants. Given the hidden and highly stigmatized nature of ARMSWs, this approach and sample size was considered appropriate for maximizing safety, trust, and participation. Similar studies involving marginalized key populations have employed comparable sample sizes to explore behavioral and awareness patterns [24]. Recruitment began with two [2] trained peer-research assistants, both ARMSWs and long-standing members of the Pink Refugees network. These peers activated their informal social and sexual networks to identify and recruit participants via WhatsApp messages and phone calls. Referrals were informal and non-incentivized, and while some participants further referred others, the majority of recruitment occurred within the network of the initial seeds.

Although respondent-driven sampling (RDS) has been validated for recruiting hidden populations, its implementation was not feasible in this study due to the small and highly interconnected nature of ARMSW networks, safety concerns, the absence of stable venues for controlled coupon-based recruitment, and ethical considerations related to incentivized peer recruitment among a vulnerable refugee population. As such, referrals were informal and non-incentivized, and recruitment remained largely within the networks of the initial peer seeds. No fixed recruitment quotas were established, and the number of individuals approached or who declined participation could not be systematically tracked. Consequently, this sampling strategy may have introduced selection bias and limited the representativeness of the sample, which was clustered within well-connected social networks.

For the qualitative component, we conducted 20 in-depth interviews (IDIs) and 2 focus group discussions (FGDs), involving a total of 35 participants. Recruitment followed similar peer-driven methods. Ethical safeguards, including verbal and written informed consent, were implemented throughout.

### Data collection procedures

During the qualitative phase, we carried out 20 in-depth interviews (IDIs) and two focus group discussions (FGDs) with ARMSWs between September and October 2023 to explore their experiences, barriers, and facilitators related to HIV testing and awareness of HIVST. To guide data collection, a semi-structured interview and FGD guide was developed based on relevant literature, expert consultation, and preliminary field observations. The IDIs lasted approximately

45–60 minutes, while each FGD lasted between 90 and 120 minutes. All FGDs and IDIs were conducted face-to-face at Pink Refugees and its affiliated locations, audio-recorded with consent, and complemented by comprehensive field notes. Insights from the qualitative component informed the creation and distribution of a structured survey to 150 ARMSWs in December 2023, evaluating the broader applicability of the qualitative findings regarding awareness of HIVST. We utilized RedCap, a secure mobile survey platform, to gather quantitative data. The survey link was shared online with participants after the research assistants explained the purpose of the survey and the steps involved in completing it. The survey was designed to minimize missing data through mandatory responses in RedCap for all key variables. All study tools (both qualitative and quantitative) were pretested among a small sample of ARMSWs (n = 2 for qualitative interviews and n = 5 for the quantitative survey) to ensure clarity, cultural relevance, and appropriateness. Feedback from the pretesting was used to refine the instruments prior to full data collection. Due to the concealed nature of RMSWs, we adopted a venue-based and snowball recruitment strategy through Pink Refugees. Two trained peer-research assistants, who were also ARMSWs, facilitated recruitment by identifying and engaging their peers during weekly meetings at Pink Refugees. Additional participants were recruited through referrals from FGD participants for IDIs and online surveys.

## Study variables

The primary outcome was awareness of oral HIVST, defined as having ever heard of HIV self-testing. In the qualitative component, participants watched a short video demonstrating the OraQuick oral HIVST procedure before being asked: "*Before today, have you seen or heard of this type of HIV testing?*" Responses were followed by open-ended questions exploring awareness, perceived benefits, and barriers.

In the survey, HIVST awareness was assessed using the question: "Have you ever heard of an HIV self-test?" (Yes/No). Additional survey variables included sociodemographic characteristics, healthcare access, HIV/STI testing behavior, and sex work-related practices. Full variable descriptions and codings are detailed in S1 Table.

## Data analysis

Descriptive statistics were used to summarize the socio-demographic characteristics of respondents, reported as frequencies and percentages for categorical variables and means with standard deviations or medians with interquartile ranges for continuous variables. Awareness of HIVST was dichotomized as 1 (aware) and 0 (not aware). An initial chi-square analysis was performed to examine the relationship between awareness of HIVST, and the independent variables. Prior to conducting chi-square analyses, assumptions were assessed by examining expected cell counts; Fisher's exact tests were used where expected counts were less than five. Variables that showed statistical significance (<0.05) in the chi-square analysis were then included in a multiple logistic regression model to explore their relationship with Awareness of HIVST. The model was specified a priori to adjust for potential confounding among sociodemographic characteristics, healthcare access variables, and HIV/STI testing history. Multicollinearity among independent variables was assessed using variance inflation factors (VIFs), with no evidence of problematic collinearity observed. Adjusted odds ratios (aORs) with 95% confidence intervals (CIs) were reported, and statistical significance was set at $p < 0.05$. All quantitative analyses were conducted using Stata (version 18).

For qualitative data, we used conventional content analysis to explore perceptions and experiences related to HIVST, following an inductive, data-driven approach appropriate for exploratory designs [25]. Transcripts from IDIs and FGDs were reviewed independently by two researchers, who identified and coded recurring ideas related to HIVST awareness using Nvivo 14. Each analyst developed a preliminary thematic summary for each transcript. The researchers then met to compare and consolidate their summaries, resolving discrepancies through discussion. The lead author reviewed all merged summaries, refined the codes, and organized emerging themes based on the study's research questions to contextualize the quantitative findings.

### Ethical approval

Ethical approval for this study was granted by the National Ethics Committee for Clinical Trials under the Italian Ministry of Health (AOO-ISS – 04/07/2023–0031228). As part of the process, the senior author and Principal Investigator presented the study during a special session with the Italian ethics committee, received immediate feedback, and provided a commitment to participant protection, including obtaining written or oral informed consent and collecting and transferring anonymous data for vulnerable populations. Hence, participants' identifiers were not linked to their data, especially in the qualitative parts of the study. Written informed consent was sort from all IDI and FGD participants before data collection commenced. Participation in the survey was voluntary, and informed consent was implied through the completion and submission of the survey after participants reviewed an online information sheet outlining the study's purpose, confidentiality, and their rights. Participants were informed that they could decline or withdraw from the study at any time without any consequences. Individuals who chose not to participate received the same services, referrals, and support provided by the partner organization as study participants.

## Results

### Sociodemographic characteristics of respondents

A total of 150 male sex workers participated in the study. The mean age was 30.6 years (SD = 5.9), with the average age at which they commenced sex work being 23.3 years (SD = 4.6). The majority (96.67%, n = 145) identified as men. More than half (57.33%, n = 97) were unmarried, and 49% (n = 74) completed senior high school or vocational training. Most participants (87.33%, n = 131) had official refugee status, and 57.33% (n = 86) originated from Nigeria. Full demographic details are provided in Table 1. In addition to the survey, 35 ARMSWs participated in the qualitative phase of the study. Participants in the IDIs had an average age of 28 years, while the 15 participants in the FGDs had an average age of 32.1 years. Participants were primarily from Nigeria and Cameroon, with a few from Côte d'Ivoire.

### Awareness of Oral HIVST among ARMSWs

**Quantitative findings on awareness of Oral HIVST among ARMSWs.** Less than half of participants (45%) were aware of HIVST (Fig 1). Among those aware, 47.8% reported ever using a self-test (Fig 2). Among participants who had used HIVST, most obtained the kit through LGBTQ+ community centers, followed by pharmacies, hospitals, and sexual health clinics (Fig 3).

**Qualitative findings on awareness of Oral HIVST among ARMSWs.** Participants' awareness of HIVST varied, with many encountering it for the first time through engagement with the study team, while others had prior exposure to it in different settings. The findings are categorized into two themes: First-time Exposure to HIVST and Prior Awareness of HIVST.

**First-time exposure to HIVST.** For several participants, this study was their first introduction to HIVST. Through a demonstration video, the study introduced the concept to them, and they found it novel, innovative, and intriguing. Some participants found the video demonstration particularly helpful, as it clarified the testing procedure and expressed excitement about its ease of use and potential benefits, such as privacy and convenience.

*Wow, I didn't know about this before. This is the first time I'm hearing about it, and I believe it's something new and innovative. It's really interesting to learn about, and I think it could be very useful – IDI participant 07*

*Before now, I had never heard about oral HIVST. It was only when I came across this program that I learned there is a way to test for HIV by yourself without even going to the hospital, which I find interesting and would like to know more about. – IDI participant 02*

**Table 1.  Sociodemographic characteristics of study participants (N = 150).**

| Variable | n (%) or Mean (SD) |
|---|---|
| **Continuous variables** | |
| Age (years) | 30.6 (5.9) |
| Age at starting sex work (years) | 23.3 (4.6) |
| Monthly income (€) | 253.5 (212.4) |
| **Categorical variables** | |
| **Education** | |
| No formal education | 7 (4.7) |
| Primary or less | 30 (20.0) |
| Senior high school/vocational | 74 (49.0) |
| Tertiary | 39 (26.0) |
| **Gender identity** | |
| Man | 145 (96.7) |
| Transgender | 3 (2.0) |
| Non-binary | 2 (1.3) |
| **Marital status** | |
| Married | 53 (35.3) |
| Single | 86 (57.3) |
| Divorced | 10 (6.7) |
| Widowed | 1 (0.7) |
| **Number of children** | |
| 0 | 80 (53.3) |
| 1 | 25 (16.7) |
| 2 | 26 (17.3) |
| 3 | 13 (8.7) |
| 4 | 6 (4.0) |
| **Religious affiliation** | |
| Christianity | 71 (47.3) |
| Islam | 37 (24.7) |
| Judaism | 1 (0.7) |
| African Traditional Religion | 16 (10.7) |
| No religion | 25 (16.7) |
| **Rank of religiosity** | |
| Extremely religious | 8 (5.3) |
| Very religious | 15 (10.0) |
| Moderately religious | 25 (16.7) |
| Maybe religious | 32 (21.3) |
| Not very religious | 55 (36.7) |
| Not religious | 15 (10.0) |
| **Immigration status** | |
| Refugee | 131 (87.3) |
| Subsidiary | 6 (4.0) |
| Motivo Specially | 8 (5.3) |
| Don't know | 5 (3.3) |
| **Length of stay in Italy** | |
| Less than 3 months | 10 (6.7) |
| 3–6 months | 21 (14.0) |

*(Continued)*

**Table 1.** (Continued)

| Variable | n (%) or Mean (SD) |
| --- | --- |
| 6 months – 1 year | 32 (21.3) |
| 1–2 years | 30 (20.0) |
| More than 2 years | 57 (38.0) |
| **Country of origin** | |
| Nigeria | 86 (57.3) |
| Cameroon | 6 (4.0) |
| Ghana | 16 (10.7) |
| Others | 42 (28.0)* |
| **Sexual orientation** | |
| Gay | 106 (70.7) |
| Bisexual | 44 (29.3) |
| **Sexual role** | |
| Bottom | 28 (18.7) |
| Top | 63 (42.0) |
| Versatile | 59 (39.3) |
| **Other job besides sex work** | |
| Yes | 32 (21.3) |
| No | 118 (78.7) |
| **Average Daily Clients** | |
| 1–5 daily clients | 133 (88.7) |
| 6–10 daily clients | 17 (11.3) |

*Note: "Others" includes countries with n < 5 participants. Full breakdown in S2 Table.

Awareness of Oral HIV Self-Testing among ARMSWs (n = 150)

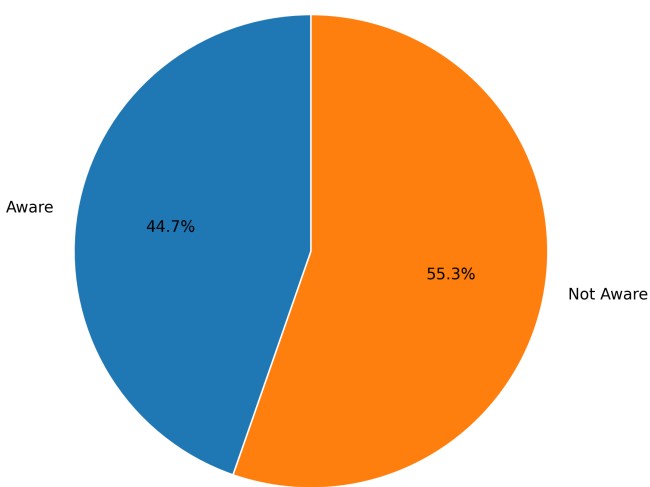

**Fig 1. Awareness of HIVST among ARMSWs (N = 150).**

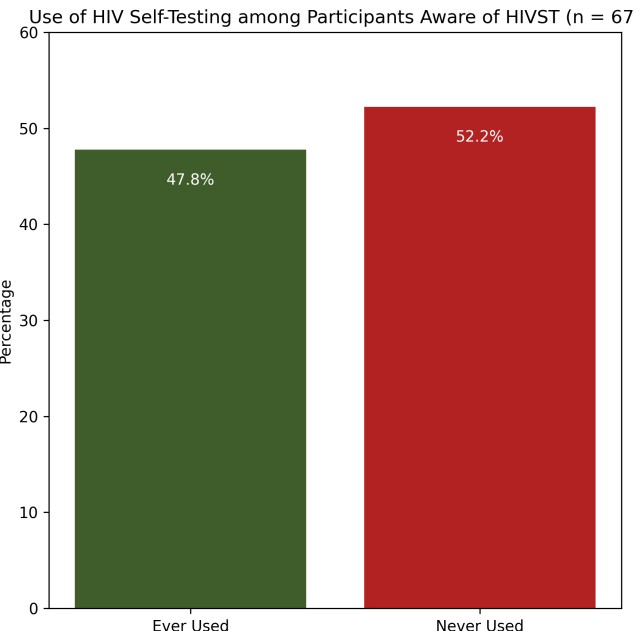

**Fig 2. Use of HIVST among Participants Aware of HIVST (N = 67).**

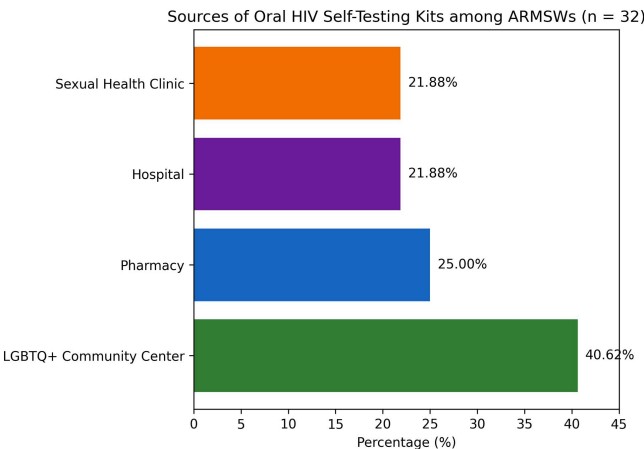

**Fig 3. Sources of HIVST Kits among ARMSWs who reported using HIVST (N = 32).**

*Yeah, you people showed me a video on the way they used to test for HIV, and it's very easy. I love it like that. It's very easy. I've never seen this kind of video before. That is my first time, and it's very good to grab that kind of opportunity… [I nod] Yeah, it's very easy. You put it in your mouth; you can just test it yourself. You don't have to worry, and you don't have to be going to the doctor too, you do it yourself. – IDI participant 03*

*No, actually no. Before you showed me the video, I had not heard of or seen this kind of testing before. – IDI participant 05*

**Prior awareness of HIVST.** While many participants encountered HIVST for the first time through the study, a few had prior knowledge or experience with self-testing. Some had seen it used by clients or peers, while others had been trained to use it through peer-led community-based programs and interventions in their home countries.

**Encounters with HIVST through clients or peers.** Some participants had seen HIVST kits used by clients or acquaintances in Italy and their home countries. They found the technology beneficial, particularly for its privacy and convenience.

*Yes, yes, there was a client….in Nigeria... I love what he did; he first asked me, Are you HIV-free?' And I said 'yes', he said 'to be on a safer path, can we run a test?' I said, 'Oh, we have to go to the hospital to…?' he said, 'No, I have my kits.' he went into the room, brought the kits, and we did the test. For me, it was very safe, private, and convenient. – IDI participant 01*

*"Yes, I saw it at a client's house here in Italy. It should be made available for all, and if possible, it should be free so most people who are ignorant about their health might see it as an advantage too, as I know myself. It saves time, it's convenient, and it's confidential." – FGD participant 03*

**Encounters with HIVST through community-based training programs.** A few participants had been exposed to HIVST through community-based training programs in their home country. Some had even been trained to use and distribute the kits to their peers in their communities.

*I have—yes, I have. Like I said, when I was in Nigeria, I worked with a peer-led group. I had friends who trained us at home on how to use these self-testing kits. We received the kits from an organization called 'The Population Council in Lagos and brought them home. Our house served as a community center where people came and went, offering support to one another. Because of this, our home became a place where self-testing kits were introduced and used. The training helped us understand how to use them properly. Before this, I didn't know where to find these kits, but through that experience, I learned a lot about self-testing. – FGD participant 02*

### Factors associated with awareness of Oral HIVST among ARMSWs

**Bivariate analysis of factors associated with awareness of Oral HIVST among ARMSWs.** Table 2 presents chi-square test results examining associations between sociodemographic factors, healthcare access, testing behavior, sex work-related characteristics, and awareness of HIVST among ARMSWs. Awareness was significantly higher among participants with secondary or higher education, those who had accessed healthcare or STI services, and those who reported facing immigration-related barriers or stigma in healthcare settings. Several sex work–related factors, including meeting clients online, having more male clients, engaging in condomless transactional sex, and inconsistent condom or lubricant use, were also significantly associated with HIVST awareness among ARMSWs.

**Univariate and multivariate logistic regression analysis of factors influencing awareness of HIVST among ARMSWs.** Table 3 presents the results of the unadjusted and adjusted logistic regression models. In the unadjusted analysis, multiple factors were significantly associated with HIVST awareness, including higher education, recent healthcare access, immigration-related barriers, STI testing, and several sex work–related behaviors such as condomless transactional sex and number of male clients. However, in the adjusted model, only education level and prior STI testing remained significant predictors of awareness. Full model results are provided in Table 3.

## Discussion

This study provides novel evidence on awareness of oral HIV self-testing among African refugee male sex workers in Italy, a population that remains largely invisible in European HIV research. Fewer than half of participants were aware of HIV

**Table 2. Chi-square analysis of factors associated with awareness of Oral HIVST among ARMSWs.**

| Variable | Awareness of HIVST (n = 150) | | |
|---|---|---|---|
| | Yes (n = 67, (44.7%) | No (n = 83, (55.3%) | P-value |
| **Sociodemographic Factors** | | | |
| **Age** | | | 0.857 |
| Young Adult (18–24) | 9 (42.86) | 12 (57.14) | |
| Adult (25+) | 58 (44.96) | 71 (55.04) | |
| **Education** | | | **0.013*** |
| Primary or less | 10 (27.03) | 27 (72.97) | |
| Secondary or more | 57 (50.44) | 56 (49.56) | |
| **Gender Identity** | | | 0.266 |
| Man | 64 (44.14) | 81 (55.86) | |
| Transgender | 1 (33.33) | 2 (66.67) | |
| non-binary | 2 (100.00) | 0 (0.00) | |
| **Marital Status** | | | 0.565 |
| Unmarried | 22 (41.51) | 31 (58.49) | |
| Married | 45 (46.39) | 52 (53.61) | |
| **Number of Children** | | | 0.160 |
| No children | 40 (50.00) | 40 (50.00) | |
| One or more children | 27 (38.57) | 43 (61.43) | |
| **Religious Affiliation** | | | 0.163 |
| No religion | 8 (32.00) | 17 (68.00) | |
| Belong to a religion | 59 (47.20) | 66 (52.80) | |
| **Rank of Religiosity** | | | 0.071 |
| Not religious | 10 (66.67) | 5 (33.33) | |
| Religious | 57 (42.22) | 78 (57.78) | |
| **Length of Stay in Italy** | | | 0.476 |
| A year or less | 26 (41.27) | 37 (58.73) | |
| More than a year | 41 (47.13) | 46 (52.87) | |
| **Sexual Orientation** | | | 0.814 |
| Gay | 48 (45.28) | 58 (54.72) | |
| Bisexual | 19 (43.18) | 25 (56.82) | |
| **What role do you play during sex?** | | | 0.499 |
| Top | 28 (44.44) | 35 (55.56) | |
| Bottom | 10 (35.71) | 18 (64.29) | |
| Versatile (Verse) | 29 (49.15) | 30 (50.85) | |
| **Any other job apart from sex work** | | | 0.604 |
| Yes | 13 (40.62) | 19 (59.38) | |
| No | 54 (45.76) | 64 (54.24) | |
| **Healthcare Access, HIV/STI Testing Behavior** | | | |
| **Distance to Healthcare Facility (Km)** | | | 0.647 |
| 1–5 km | 34 (46.58) | 39 (53.42) | |
| 6 km or more | 33 (42.86) | 44 (57.14) | |
| **Healthcare frequency** | | | **0.004*** |
| Never | 6 (20.69) | 23 (79.31) | |
| Have ever | 61 (50.41) | 60 (49.59) | |

*(Continued)*

**Table 2.** (Continued)

| Variable | Awareness of HIVST (n = 150) | | |
|---|---|---|---|
| | Yes (n = 67, (44.7%) | No (n = 83, (55.3%) | P-value |
| **Health Insurance Status** | | | 0.061 |
| Yes | 31(54.39) | 26 (45.61) | |
| No | 36(38.71) | 57 (61.29) | |
| **Immigration challenge in healthcare access** | | | **0.024*** |
| Yes | 34 (55.74) | 27 (44.26) | |
| No | 33 (37.08) | 56 (62.92) | |
| **Experienced stigma or discrimination at the hospital** | | | **0.012*** |
| Yes | 36 (37.11) | 61(62.89) | |
| No | 31(58.49) | 22 (41.51) | |
| **Do you know where to go for STI/HIV testing and care?** | | | **0.000*** |
| Yes | 54 (58.06) | 39 (41.94) | |
| No | 13 (22.81) | 44 (77.19) | |
| **Tested for STI** | | | **0.000*** |
| Never tested | 10 (22.22) | 35(77.78) | |
| Ever tested | 57 (54.29) | 48 (45.71) | |
| **Tested for HIV** | | | 0.598 |
| Never tested | 27 (42.19) | 37 (57.81) | |
| Ever tested | 40 (46.51) | 46 (53.49) | |
| **HIV test frequency** | | | 0.540 |
| Once a year or less | 18 (50.00) | 18 (50.00) | |
| More than once a year | 23 (43.40) | 30 (56.60) | |
| **HIV test result** | | | 0.908 |
| Negative | 49 (44.95) | 60 (55.05) | |
| Positive | 18 (43.90) | 23 (56.10) | |
| **Sex Work–Related Factors** | | | |
| **Venue for meeting client** | | | **0.022*** |
| Offline | 66 (47.14) | 74 (52.86) | |
| Online | 1 (10.00) | 9 (90.00) | |
| **Average Daily Clients** | | | 0.25 |
| 1–5 daily clients | 60 (45.1) | 73 (54.9) | |
| 6–10 daily clients | 7 (41.2) | 10 (58.8) | |
| **No. of client in past 2month (Men)** | | | **<0.001*** |
| Less than 20 | 48 (58.54) | 34 (41.46) | |
| 20 or more | 19 (27.94) | 49 (72.06) | |
| **No. of client in past 2month (Women)** | | | 0.496 |
| Less than 10 | 19 (46.34) | 22 (53.66) | |
| 10 or more | 1 (33.33) | 2 (66.67) | |
| **Vaginal sex** | | | 0.227 |
| Yes | 44(41.51) | 62 (58.49) | |
| No | 23 (52.27) | 21 (47.73) | |
| **Condom use for vaginal sex** | | | 0.167 |
| Sometimes/Never | 38 (57.58) | 28 (42.42) | |
| Always | 8 (40.00) | 12 (60.00) | |

*(Continued)*

**Table 2.** (Continued)

| Variable | Awareness of HIVST (n = 150) | | P-value |
|---|---|---|---|
| | Yes (n = 67, (44.7%) | No (n = 83, (55.3%) | |
| **Anal sex** | | | 0.142 |
| Yes | 52 (41.94) | 72 (58.06) | |
| No | 15 (57.69) | 11 (42.31) | |
| **Condom use for anal sex** | | | **0.029*** |
| Sometimes/Never | 49 (41.53) | 69 (58.47) | |
| Always | 13 (68.42) | 6 (31.58) | |
| **Lubricant use for anal sex** | | | **0.037*** |
| Yes | 40 (39.60) | 61 (60.40) | |
| No | 21 (60.00) | 14 (40.00) | |
| **Transactional condomless sex** | | | **<0.001*** |
| No | 21 (29.17) | 51(70.83) | |
| Yes | 46 (58.97) | 32(41.03) | |

self-testing, and among those aware, less than half had ever used a self-test. These findings underscore a substantial gap between the availability of HIVST in Italy and its reach among one of the most marginalized populations at risk of HIV infection.

The observed level of HIVST awareness is consistent with European studies among MSM and migrant populations that report persistently low awareness and uptake several years after HIVST authorization [19–22]. However, given the compounded vulnerabilities faced by ARMSWs, including migration-related precarity, stigma, and limited engagement with healthcare systems, the low awareness identified in this study is particularly concerning. These findings suggest that current HIVST dissemination strategies in Italy may not adequately reach populations positioned at the margins of both migration and sex work.

Higher educational attainment emerged as a significant predictor of HIVST awareness, aligning with broader evidence linking education to health literacy and engagement with preventive health services [23,26,27]. Among ARMSWs, education may facilitate navigation of health information, interpretation of testing instructions, and awareness of novel prevention tools. This finding highlights the need for HIVST promotion strategies that account for varying literacy levels and language barriers, including the use of visual aids, peer demonstrations, and culturally adapted materials.

Prior engagement with healthcare services, particularly STI testing, was also strongly associated with HIVST awareness. This reinforces the importance of healthcare encounters as key points for information dissemination, even among populations that experience barriers to routine care. Integrating HIVST education into STI testing services and community-based sexual health programs could therefore represent a practical strategy for increasing awareness among ARMSWs. [28]

Qualitative findings further revealed that awareness pathways were often informal and unintentional, with some participants first encountering HIVST kits in clients' homes rather than through health services or outreach programs. This underscores the role of social and sexual networks in shaping access to HIV prevention information and highlights missed opportunities for structured, targeted promotion [29]. Participants who learned about HIVST expressed strong interest in its privacy, convenience, and autonomy, attributes that are particularly salient for individuals who fear stigma, disclosure, or discrimination in healthcare settings.[30].

Several structural and behavioral factors, including immigration-related barriers to healthcare, experiences of stigma in hospitals, and sex work practices, were associated with HIVST awareness in bivariate analyses but did not retain

**Table 3. Crude (unadjusted) and adjusted multiple logistic regression analysis of factors influencing the awareness of HIVST among ARMSWs.**

| Awareness of HIVST | Unadjusted Model | | Adjusted Model | |
|---|---|---|---|---|
| | cOR (95% CI) | P-value | aOR (95% CI) | P-value |
| **Sociodemographic Factors** | | | | |
| **Education** | | | | |
| Primary or less | Ref | Ref | Ref | Ref |
| Secondary or higher | 2.75 (1.22–6.20) | **0.015*** | 1.92 (0.70–5.29) | **0.022*** |
| **Healthcare Access, HIV/STI Testing Behavior** | | | | |
| **Healthcare frequency** | | | | |
| Never **(Ref)** | **cOR (95% CI)** | **P-value** | **aOR (95% CI)** | **P-value** |
| Have ever | 3.10 (1.48–10.25) | **0.006*** | 2.12(0.52–8.62) | 0.291 |
| **Immigration challenge in healthcare access** | | | | |
| No | Ref | Ref | Ref | Ref |
| Yes | 2.13 (1.10–4.15) | **0.025*** | 1.93 (0.87–4.28) | 0.106 |
| **Experienced stigma or discrimination at the hospital** | | | | |
| No | Ref | Ref | Ref | Ref |
| Yes | 2.09 (1.21–3.60) | **0.008*** | 1.47 (0.74–2.90) | 0.274 |
| **Do you know where to go for STI/HIV testing and care?** | | | | |
| No | Ref | Ref | Ref | Ref |
| Yes | 6.21 (5.10–7.45) | **0.000*** | 0.49 (0.188–1.27) | 0.143 |
| **Tested for STI** | | | | |
| Never tested | Ref | Ref | Ref | Ref |
| Ever tested | 4.16 (1.87 - 9.26) | **0.000*** | 2.32 (0.82–6.62) | **0.015*** |
| **Sex Work–Related Factors** | | | | |
| **No. of client in past 2month (Men)** | | | | |
| Less than 20 | Ref | Ref | Ref | Ref |
| 20 or more | 1.97 (0.94–2.69) | **<0.001*** | 0.52 (0.22 - 1.25) | 0.144 |
| **Condom use for anal sex** | | | | |
| Sometimes/Never | Ref | Ref | Ref | Ref |
| Always | 3.05 (1.08–8.58) | **0.035*** | 1.24 (0.74–2.09) | 0.418 |
| **Lubricant use for anal sex** | | | | |
| No | Ref | Ref | Ref | Ref |
| Yes | 2.29 (1.04–5.02) | **0.039*** | 1.49 (0.58–3.81) | 0.403 |
| **Transactional condomless sex** | | | | |
| No | Ref | Ref | Ref | Ref |
| Yes | 3.49 (1.77–6.89) | **<0.001*** | 0.98 (0.39–2.42) | 0.96 |

cOR (Crude odds ratio); aOR(Adjusted odds ratio); 95% CI – (95% Confidence interval); *(P ≤ 0.05).

significance in the adjusted model. While these findings suggest confounding by education and healthcare engagement, they nonetheless point to the broader social context shaping awareness. Experiences of exclusion from formal health systems may indirectly drive interest in alternative testing approaches such as HIVST, even if these factors do not independently predict awareness. [16,31]

Taken together, these findings have important implications for HIV prevention efforts targeting ARMSWs in Italy. Expanding HIVST awareness will require deliberate, community-centered strategies that move beyond passive availability. Peer-led interventions, delivered through trusted community organizations, may be particularly effective given

the importance of social networks observed in this study [32–34]. Additionally, engaging clients of sex workers in HIVST promotion could represent an underexplored avenue for normalizing testing and reducing transmission risk within sexual networks.

### Study strengths and limitations

A major strength of the study is the mixed-methods design, which enabled integration of quantitative patterns with in-depth qualitative narratives, offering a nuanced understanding of awareness pathways, perceptions, and associated factors. Peer-led recruitment facilitated trust and access to a hidden and stigmatized population that is often excluded from routine HIV surveillance. Conducting the study across two cities in Northern Italy allowed for some contextual variation, and triangulation of survey and interview data strengthened the credibility of the findings.

Several limitations should be acknowledged. All data were self-reported and may be subject to recall and social desirability bias, particularly for sensitive behaviors related to HIV testing and sex work. Quantitative data on fees charged per client were not collected. Qualitative findings indicated that pricing among ARMSWs was highly variable and negotiated on a case-by-case basis, influenced by factors such as client characteristics, venue, duration of the encounter, perceived risk, and participants' personal preferences. As a result, a single numeric measure of charges per client would not have adequately captured economic vulnerability and may have oversimplified complex pricing dynamics. The cross-sectional design precludes causal inference, and observed associations should be interpreted as correlational. Additionally, the peer-led venue-based and snowball sampling approach, while ethically appropriate and pragmatic, may have introduced selection bias, with recruitment likely concentrated within well-connected social networks. The modest sample size and focus on Verona and Turin further limit generalizability to other regions in Italy or Europe. Despite these limitations, this study represents one of the first empirical examinations of HIVST awareness among ARMSWs in Italy and provides an important foundation for future research and targeted HIV prevention efforts.

### Conclusion

Our findings highlight the urgent need for targeted HIVST awareness initiatives among ARMSWs in Italy. Education and prior STI testing play key roles in awareness, suggesting that interventions should focus on health literacy and engagement with healthcare services. Community-based outreach, peer-led education, and healthcare-provider involvement are critical strategies for improving uptake. Future research should explore long-term behavioral changes following HIVST promotion efforts to assess their impact on testing rates and linkage to care.

### Supporting information

**S1 Table. Study variables and coding scheme used in the quantitative analysis.**
(DOCX)

**S2 Table. Country of origin of African refugee male sex workers included in the study (n = 150).**
(DOCX)

### Author contributions

**Conceptualization:** Gamji Rabiu Abu-Ba'are, Giovanni Zardini.

**Data curation:** Gamji Rabiu Abu-Ba'are, Giovanni Zardini.

**Formal analysis:** Henry Delali Dakpui, Mubarik Sena Saaka.

**Funding acquisition:** Gamji Rabiu Abu-Ba'are.

**Investigation:** Gamji Rabiu Abu-Ba'are, Giovanni Zardini.

**Methodology:** Gamji Rabiu Abu-Ba'are, Donte Boyd, Orlando Harris.

**Project administration:** Gamji Rabiu Abu-Ba'are.

**Supervision:** Gamji Rabiu Abu-Ba'are, Giovanni Zardini.

**Validation:** Donte Boyd.

**Writing – original draft:** Henry Delali Dakpui, Mubarik Sena Saaka, Donte Boyd, Orlando Harris.

**Writing – review & editing:** Henry Delali Dakpui, Gamji Rabiu Abu-Ba'are, Mubarik Sena Saaka, Donte Boyd, Orlando Harris, Giovanni Zardini.

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
