## [Decision Letter · Decision Letter 0]

30 Dec 2025

Dear Dr. Abu-Ba'are,

We look forward to receiving your revised manuscript.

Kind regards,

Serge Tonen-Wolyec, M.D., Ph.D.

Academic Editor

PLOS One

Journal Requirements:

2. Please provide additional information regarding the considerations  made for the Refugee included in this study. For instance, please discuss whether participants were able to opt out of the study and whether individuals who did not participate receive the same treatment offered to participants.

3. Please include a complete copy of PLOS’ questionnaire on inclusivity in global research in your revised manuscript. Our policy for research in this area aims to improve transparency in the reporting of research performed outside of researchers’ own country or community. The policy applies to researchers who have travelled to a different country to conduct research, research with Indigenous populations or their lands, and research on cultural artefacts. The questionnaire can also be requested at the journal’s discretion for any other submissions, even if these conditions are not met. Please find more information on the policy and a link to download a blank copy of the questionnaire here: https://journals.plos.org/plosone/s/best-practices-in-research-reporting. Please upload a completed version of your questionnaire as Supporting Information when you resubmit your manuscript.

“The project described was supported by award number P30MH062294 from the National Institute of Mental Health (NIMH) through the Center for Interdisciplinary Research on AIDS (CIRA), Yale School of Public Health. The content is solely the responsibility of the authors and does not necessarily represent the official views of the Center for Interdisciplinary Research on AIDS, the National Institute of Mental Health, or the National Institutes of Health “

5. We note that you have indicated that there are restrictions to data sharing for this study. For studies involving human research participant data or other sensitive data, we encourage authors to share de-identified or anonymized data. However, when data cannot be publicly shared for ethical reasons, we allow authors to make their data sets available upon request. For information on unacceptable data access restrictions, please see http://journals.plos.org/plosone/s/data-availability#loc-unacceptable-data-access-restrictions.

6. We notice that your supplementary tables are included in the manuscript file. Please remove them and upload them with the file type 'Supporting Information'. Please ensure that each Supporting Information file has a legend listed in the manuscript after the references list.

Additional Editor Comments:

The manuscript is very well written, with details that are important for understanding the methodology and the main findings.

The major weakness of this article is the use of non-probability sampling for the quantitative survey phase. I would have liked to see the use of sampling methods that have already been validated for hard-to-reach populations, such as respondent-driven sampling (RDS).

Statistically, the authors should specify that the validity of the chi-square tests was systematically verified and how they controlled for confounding factors.

Reviewers' comments:

Reviewer's Responses to Questions

**Comments to the Author**

1. Is the manuscript technically sound, and do the data support the conclusions?

Reviewer #1: Yes

Reviewer #2: Yes

2. Has the statistical analysis been performed appropriately and rigorously?

Reviewer #1: Yes

Reviewer #2: Yes

3. Have the authors made all data underlying the findings in their manuscript fully available?

Reviewer #1: Yes

Reviewer #2: Yes

4. Is the manuscript presented in an intelligible fashion and written in standard English?

Reviewer #1: Yes

Reviewer #2: Yes

Reviewer #1: Dear authors

Thank you for the opportunity to review your manuscript: Factors Associated with Oral HIV Self-Testing Awareness Among African Refugee Male Sex Workers in Italy. A Mixed-Methods Study. I have carefully completed the revision and provided detailed feedback below, this manuscript examines HIVST awareness and associated factors among ARMSWs to inform targeted interventions.

Congratulations, this is a well written manuscript.

1. Please revise your title to make it punchier and impactful.

2. Please revise your introduction, to make it shorter and I would suggest you focus on research gaps regarding ARMSWs in Europe.

3. Please repeat table header on subsequent pages.

4. Please do you have information on how much participants charge per client? I would suggest you to include it to highlight their vulnerability as they earn 250euros/month mean (use range best) and 58% see less than 20 clients per month.

5. Please include a new "Sample and Sampling" subsection to better describe the peer-referral (snowball) recruitment process.

6. Please avoid ending your introduction with questions, I would suggest rephrasing the objective in the last paragraph as text.

7. Please further condense some of the information in your discussion and focus on interpreting your key findings (while making sure they tie back to what was presented in the introduction section), to make the paper more cohesive, and please include more information on what would be the implications for future research and clinical practice as ARMSWs are a very particular population in Italy : very vulnerable, small size, often without a regulated immigration status, language barriers, etc.

8. Please make sure to revise the references and typos in general .

9. Please make sure to revise and maybe cite other publications for HIVST in Italy/Europe, for example: https://pubmed.ncbi.nlm.nih.gov/39863901/, https://pubmed.ncbi.nlm.nih.gov/35784245/,
https://pubmed.ncbi.nlm.nih.gov/25394102/

Reviewer #2: The paper has been extensively and correctly revised according to all issues raised by the referees, in particular the methodological issues and the addition of a full "Study Strengths and Limitations" paragraph. The paper is relevant on a topic rarely addressed in medical literature.

**Do you want your identity to be public for this peer review?** For information about this choice, including consent withdrawal, please see our Privacy Policy

Reviewer #1: **Yes:** CARMEN FIGUEROA

Reviewer #2: No

---

## [Author Response · Author response to Decision Letter 1]

23 Jan 2026

We have carefully addressed all comments raised by the Academic Editor and reviewers. Revisions include clarification of the sampling approach and statistical analyses, revision and condensation of the Introduction and Discussion, updates to tables, figures, and supporting information, and clarification of ethical considerations, data availability, and funding disclosure. A detailed, point-by-point Response to Reviewers is provided in the uploaded document.

---

## [Editor Report · Decision Letter 1]

5 Feb 2026

HIV Self-Testing Awareness Among African Refugee Male Sex Workers in Italy. A Mixed-Methods Study

PONE-D-25-61075R1

Dear Dr. Abu-Ba’are,

We’re pleased to inform you that your manuscript has been judged scientifically suitable for publication and will be formally accepted for publication once it meets all outstanding technical requirements.

Kind regards,

Serge Tonen-Wolyec, M.D., Ph.D.

Academic Editor

PLOS One

---

## [Editor Report · Acceptance letter]

PONE-D-25-61075R1

PLOS One

Dear Dr. Abu-Ba'are,

I'm pleased to inform you that your manuscript has been deemed suitable for publication in PLOS One. Congratulations! Your manuscript is now being handed over to our production team.

Kind regards,

on behalf of

Dr. Serge Tonen-Wolyec

Academic Editor

PLOS One